# Deciphering Conformational Changes of the GDP-Bound NRAS Induced by Mutations G13D, Q61R, and C118S through Gaussian Accelerated Molecular Dynamic Simulations

**DOI:** 10.3390/molecules27175596

**Published:** 2022-08-30

**Authors:** Zhiping Yu, Hongyi Su, Jianzhong Chen, Guodong Hu

**Affiliations:** 1Shandong Key Laboratory of Biophysics, Institute of Biophysics, Dezhou University, Dezhou 253023, China; 2Laoling People’s Hospital, Dezhou 253600, China; 3School of Science, Shandong Jiaotong University, Jinan 250357, China

**Keywords:** NRAS, GDP, MR-GaMD simulations, free energy landscapes, principal component analysis

## Abstract

The conformational changes in switch domains significantly affect the activity of NRAS. Gaussian-accelerated molecular dynamics (GaMD) simulations of three separate replicas were performed to decipher the effects of G13D, Q16R, and C118S on the conformational transformation of the GDP-bound NRAS. The analyses of root-mean-square fluctuations and dynamics cross-correlation maps indicated that the structural flexibility and motion modes of the switch domains involved in the binding of NRAS to effectors are highly altered by the G13D, Q61R, and C118Smutations. The free energy landscapes (FELs) suggested that mutations induce more energetic states in NRAS than the GDP-bound WT NRAS and lead to high disorder in the switch domains. The FELs also indicated that the different numbers of sodium ions entering the GDP binding regions compensate for the changes in electrostatic environments caused by mutations, especially for G13D. The GDP–residue interactions revealed that the disorder in the switch domains was attributable to the unstable hydrogen bonds between GDP and two residues, V29 and D30. This work is expected to provide information on the energetic basis and dynamics of conformational changes in switch domains that can aid in deeply understanding the target roles of NRAS in anticancer treatment.

## 1. Introduction

Small GTPases proteins (KRAS, HRAS, and NRAS) coded by human RAS genes regulate the levels of numerous cell signaling processes implicated in cell growth [1,2,3]. KRAS, HRAS, and NRAS have in common the feature of exchanging between the GTP-bound active state and the GDP-bound inactive state [4,5,6]. According to previous reports, the hydrolysis reaction of guanosine triphosphate (GTP) into guanosine diphosphate (GDP) can be accelerated by the GTPase activating proteins (GAPs), which generate an inactive form of RAS protein [7]. In contrast, the transformation of GDP into GTP can be catalyzed by guanosine exchange factors, leading to an active state for the GTP-associated RAS protein [7,8]. Significant signaling pathways, responsible for key roles in cell proliferation and survival, are triggered by the binding of the GTP-bound RAS proteins to effectors, mainly involving RAF, phosphoinositide 3-kinase (PI3K), and Ral guanine nucleotide dissociation stimulator (RalGDS) [9]. Residue mutation-mediated active RAS proteins generally drive human cancers and tumorigenesis by hyperactivating the downstream signal pathways [10,11,12]. Therefore, exploring the effects of mutations on the activity of RAS proteins would be of great significance to for a deeper understanding of the target role of RAS proteins in human cancers.

Among human cancers induced by mutations of RAS proteins, KRAS is the most frequently mutated oncogene, mainly occurring at the residues G12 and G13 and accounting for respective mutation rates of 86–96% in pancreatic cancers [13], 40–54% in colorectal cancers [1], and 27–39% in lung adenocarcinomas [14]. Mutations of HRAS are mostly detected in salivary gland and urinary tract cancers [15,16], while those of NRAS are mainly identified in melanoma and hematological malignancies [17,18]. Furthermore, mutations from NRAS at codons 50 and 60 are also reported to be related to Noonan syndrome [19]. Presently, although there are fewer studies on human cancers caused by mutations of NRAS than on those induced by mutations of KRAS, it is known that mutationsof NRAS are indeed involved in human tumorigenesis [20,21,22]. Thus, it is essential to further investigate the molecular mechanism of NRAS mutation-mediated cancers to look for an efficient pathway of drug design for the treatment of human cancers.

NRAS shares a similar structural topology with KRAS and HRAS (Figure 1A), in which the switch domains of NRAS consist of switch I (SW I, residues 25–40) and switch II (SW II, residues 59–75) [23,24,25,26,27]. As shown in Figure 1A, SW I, SWII, and the P-loop encircle a binding pocket of GTP or GDP. The X-ray and NMR experiments by Pálfy et al. revealed that the conformations of the switch domains from the RAS proteins are highly affected by G12 mutations [28,29]. Mutations lead to large conformational changes in the two switches, SW I and SW II, which yield a significant effect on the activity of NRAS [30,31]. The family members of the RAS proteins can be exchanged between two different conformational states defined by binding to differential nucleotides (GTP and GDP), which regulate the activity of the RAS protein. The binding of the GDP nucleotide leads to an ”off” state for the RAS proteins, while the binding of the GTP nucleotide results in an ”on” state for the RAS proteins, which induces a large conformational change in the switch regions [32,33,34,35,36]. It is well-known that the conformational changes in the switch domains caused by residue mutations and inhibitor bindings play vital roles in the regulation of RAS protein activity [37,38,39,40,41]. Despite rich information on the KRAS and HRAS conformational changes, studies on the conformational alterations of NRAS are still lacking; hence, it is necessary to deeply investigate the influences of mutations on the conformational changes of NRAS to understand the target role of NRAS in drug development toward cancer treatment.

Molecular dynamics (MD) simulations [42,43,44,45,46,47,48] and MD trajectory-based post-processing analysis [49,50,51,52,53] have been used as significant technology tools to probe the conformational changes in proteins caused by residue mutations and ligand bindings. MD simulations and free energy analysis have been applied to successfully investigate the mechanism regulating the activity of RAS proteins [54,55,56,57]. Recently, accelerated molecular dynamics (aMD) [58,59] and Gaussian accelerated molecular dynamics (GaMD) [60,61] simulations were proposed to enhance the conformational samplings of proteins and efficiently overcame the possibility present for conformations sampled by conventional MD (cMD) simulations of falling into a locally minimal space. Furthermore, GaMD simulations obtained great success in providing insights into the molecular mechanisms of conformational changes in proteins [62,63,64,65,66,67]. In our previous studies, aMD and GaMD simulations were used to improve conformational samplings of HRAS and KRAS and to rationally decipher the mutation-mediated effects on the conformational alterations of these two proteins [68,69].

To decode the mutation-mediated impacts on the conformational changes of NRAS, the GDP-bound wild-type (WT), G13D, Q61R, and C118S NRAS were selected in this work. Mutation sites of G13D, Q61R, and C118S are depicted in Figure 1B, and the structure of GDP is exhibited in Figure 1C. The mutation of G13D occurs at the P-loop and brings a negative net charge, while the mutation of Q61R occurs at the SW II and yields a positive net charge, and both certainly produce significant influences on the interactions of the P-loop and the SW II with GDP. The mutation C118S is generated at an allosteric position and can affect the activity of NRAS. To achieve our aim, multiple-replica GaMD (MR-GaMD) simulations, free energy landscapes (FELs), principal component analysis (PCA) [70,71,72,73], and dynamics cross-correlation maps (DCCMs) [71,74] were adopted in the current study. This work is expected to contribute useful information on dynamics and energy basis to aid the understanding of the target role of NRAS in drug design aimed at anticancer treatment.

## 2. Results and Discussion

### 2.1. Structural Flexibility and Motion Modes between Structure Domains of NRAS

To investigate the effects of G13D, Q61R, and C118S on the local structural flexibility of NRAS, the difference in the RMSF for each C_α_ atom between the mutated and the wild-type NRAS was estimated through the equation,$\;\Delta RMSF=RMSF_{mutant}-RMSF_{WT}$ (Figure 2A). It was found that the most obvious effect of mutations on the structural flexibility occurs at the two switch domains SWI and SWII. The mutations of G13D and Q61R weaken the structural flexibility of the SWI, but C118S strengthens it (Figure 1A and Figure 2A). In contrast the weak impact of C118S on the structural flexibility of the SWII, G13D greatly enhances its structural flexibility; however, Q61R strongly reduces the structural flexibility of this switch domain (Figure 1A and Figure 2A). It was also observed that G13D and C118S slightly strengthen the structural flexibility of loop L1. Furthermore, G13D increases the structural flexibility of loops L3 and L4, while Q61R slightly weakens that of loop L3 (Figure 1A and Figure 2A).

To reveal mutation-mediated influences on the global flexibility of NRAS, molecular surface areas (MSAs) of the GDP-bound WT, G13D, Q61R, and C118S NRAS were calculated by using the single-joined MR-GaMD trajectory (SJMT), and their frequency distributions are depicted in Figure 2B. The MSAs of the GDP-bound G13D, Q61R, and C118S NRAS were decreased by 242.5, 238.5, and 235.5 Å^2^, respectively, which indicates a lower global structural flexibility for NRAS with the binding state caused by the mutations, G13D, Q61R, and C118S.

To probe mutation-induced effects on motion modes between structural domains of NRAS, DCCMs of the C_α_ atoms from the GDP-bound WT, G13D, Q61R, and C118S NRAS were calculated based on the SJMT (Figure 2C–F). It can be noted that G13D, Q61R, and C118S exert evident effects on correlated motions between structural domains of NRAS. In the GDP-bound WT NRAS (Figure 2C), the domains D1 and D2 produce slightly anti-correlated motions (cyan), while the domain D3 generates obvious anti-correlated movement (blue). Among the D1–D3 domains from the GDP-bound WT NRAS, the D1 and D2 domains, respectively, exhibit slightly anti-correlated movements for the switches SW I and SW II relative to the P-loop (residues 8–16), while the D3 domain shows strong anti-correlated motion for the switch SW II relative to the SW. The domains D4 and D5 from the GDP-bound WT NRAS yield strongly positive correlated motions, indicated by the yellow and red in Figure 2C, in which the D4 shows strongly positive correlated motion for the β-sheet β2 relative to β1 (Figure 1A and Figure 2C),while the D5 domain shows strongly positive correlated movement between the loop L4 and the P-loop (Figure 1A and Figure 2C). Compared to the GDP-bound WT NRAS, G13D obviously strengthens the anti-correlated motions of the SWI and SWII relative to the P-loop but slightly weakens the anti-correlated movement between the SWI and SWII (Figure 2D);moreover, G13D strengthens the positive correlated motions of β2 relative to β1 and ofL4 relative to the P-loop (Figure 2D). In comparison with the GDP-bound WT NRAS, Q61R weakens not only the anti-correlated motion between the SWI and the SW II but also the positive correlated motions of β2 relative to β1 and ofL4 relative to the P-loop (Figure 2E). By interacting with the GDP-bound WT NRAS, C118S slightly strengthens the anti-correlated movement between the SWI and the SW II but softly weakens the positive correlated motion of L4 relative to the P-loop (Figure 2F).

It was found from the above analyses that G13D, Q61R, and C118S change the structural flexibility and correlated motions of the switch domains. Structurally, the switch domains are involved in the binding of the RAS proteins to effectors. Therefore, the alterations in the conformation and internal dynamics of the switch domains certainly produce a significant effect on the binding of NRAS to effectors, which regulates the activity of NRAS. The work of Kessler et al. suggests that Q61R and C118S change the “on” or “off” state of NRAS and affect the binding modes of inhibitors to NRAS [30], which agrees with our current findings. The study from Johnson et al. verifies that G13D greatly changes the structural flexibility of the switch domains and exerts vital impacts on the activity of NRAS [31], supporting our results well.

### 2.2. Conformational Changes of NRAS Revealed by Free Energy Landscapes

To reveal the energetic basis of mutation-mediated conformational changes of NRAS, FELs were constructed using the distance of residue D33 from residue 61 (Q61 or R61) and root-mean-square deviations (RMSDs) of backbone atoms from NRAS as reaction coordinates. The reasons why we selected the seas reaction coordinates are as follows: (1) the distance from D33 to residue 61 efficiently reflects the transformation of the conformational state between SW I and SW II, (2) the RMSDs of backbone atoms can rationally embody the total structural fluctuations of NRAS through MR-GaMD simulations. To understand the changes in the interaction sites of GDP with NRAS, a protein−ligand interaction profiler (PLIP) server [75,76] was applied to detect interaction networks of GDP with NRAS in different energetic states.

For the GDP-bound WT NRAS, MR-GaMD simulations detected two main energy valleys EV1 and EV2 (Figure 3A), indicating that the GDP-bound WT NRAS is distributed at two conformational sub-spaces. The depth of the potential well EV1 was much deeper than that of the EV2, implying that the probability of transition from EV2 into EV1 of NRAS is much higher than that from EV1 into EV2. Thus, the conformations of the GDP-bound WT NRAS mostly fall into EV1. The distances of D33 from Q62 in the energetic states EV1 and EV2 were25.4 and 12.4 Å, respectively, showing that the switch domains SWI and SWII of the GDP-bound WT NRAS situated at EV1 have an open conformation, while those located at EV2 exist in a closed conformation (Figure 3B,C). In two energetic states, GDP forms a salt bridge interaction with K16 and D119 while GDP produces hydrogen bonding interactions (HBIs) with the common residues G15, K16, S17, A18, V29, D30, N116, K117, and K147 (Appendix A). Although GDP loses two HBIs with V14 and S146 in EV2 compared to EV1, two new HBIs also appear between GDP and the two residues G13 and A145 in EV2 (Appendix A). The representative structure at EV1 was superimposed together with that at the EV2 (Figure 3D). It was found that the switch domains SWI and II become highly disordered by interacting with the other structural domains. In addition, the helix α3 and the loops L1 and L3 generate slight deviations. Moreover, the structures of the GDP and Mg^2+^ion situated at EV1 and EV2 are aligned together (Appendix A). The adenine group and the middle ring from GDP obviously slide away from each other and the Mg^2+^ions in EV1 and EV2 generate a deviation of 2.2 Å (Figure 1C), which can strongly affect the binding of NRAS to effectors.

With regard to the GDP-bound G13D NRAS, three low energetic valleys EV1–EV3 were captured by MR-GaMD simulations (Figure 4A), showing that the GDP-bound G13D NRAS is mainly clustered into three conformational sub-spaces. The potential well EV1 was found to be deeper than EV2 and EV3; as a result, the GDP-bound G13D NRAS was mostly located at the energetic valley EV1 (Figure 4A). The distances of D33 from Q61 were15.0, 18.1, and 16.4 Å at EV1, EV2, and EV3, respectively; hence, the switch domains SW I and SWII form an open state at EV2, while the switch domains SWI and II are located at an intermediate state at EV1 and EV3 (Figure 4B–D). Similarly to the GDP-bound WT NRAS, GDP forms salt bridge interactions with K16 and D119 in three energetic states EV1, EV2, and EV3 (Appendix A). Meanwhile, GDP yields HBIs with the conserved residues G15, K16, S17, A18, N116, K117, S145, A146, and K147 (Appendix A). However, G13D also induces an alteration in interaction sites of GDP with NRAS. In detail, it was found that two new HBIs appear between GDP and the residues D13 and V29 in EV1 (Appendix A), GDP generates two additional HBIs with V14 and D33 in EV2 (Appendix A), and GDP not only forms three additional HBIs with D13, V14, and D30 but also produces a new salt bridge interaction with D30 in EV3 (Appendix A). The alignment of the structures situated at EV1, EV2, and EV3 was carried out (Figure 4E), and the results showed that the switch domains are in a highly disordered state. It was also found that two additional Na^+^ ions appear at the EV1 and a Na^+^ ion is at EV3 (Figure 4B,D,E). The structures of GDP and Mg^2+^ions in EV1–EV3 were also superimposed together (Figure 4F). The results indicated that the adenine group, the middle ring, and the phosphate group of GDP are aligned well. Although G13D induces more energetic states compared to the GDP-bound WT NRAS, the appearance of Na^+^ ions caused by G13D maintains the structural stability of GDP.

In the case of the GDP-bound Q61R NRAS, four energetic valleys EV1–EV4 were detected in the MR-GaMD simulations (Figure 5A), suggesting that the GDP-bound Q61R NRAS is mainly classed into four conformational sub-spaces. The depths of the four potential wells EV1–EV4 were almost the same; hence, the conformations of the GDP-bound Q61R NRAS were almost equally distributed at four conformational sub-spaces. The distances of D33 from R61 were 12.6, 16.2, 19.4, and 23.3 Å in the energetic states EV1, EV2, EV3, and EV4, respectively(Figure 5B–E). Therefore, it was found that the switch domains SW I and SWII form a closed state in EV1, induce an intermediate state in EV2, and lead to two open states in EV3 and EV4. In the four energetic states EV1–EV4, GDP produces two salt bridge interactions with K16 and D119 (Appendix A). It was also observed that GDP forms HBIs with the conserved residues V14, G15, K16, S17, N116, K117, S145, and K147 in all four energetic states (Appendix A). Interestingly, a π–π interaction appears between F28 and the adenine group of GDP in EV1 and EV2 (Appendix A). In EV1, EV3, and EV4, GDP generates an additional HBI with A146 (Appendix A). The residue D30 forms an HBI with the phosphate group of GDP in EV1 and EV4 (Appendix A). In addition, GDP also yields an HBI with V29 from the SWI in the EV1. The representative structures situated at EV1–EV4 were superimposed together (Figure 5F). As exhibited in Figure 5F, the switch SWI has a highly disordered state while the SWII only generates obvious deviations. In addition, the helix α3 and the loops L1 and L4 also produce evident deviations. It was also observed that additional sodium ions Na^+^ respectively appear at EV2, EV3, and EV4 (Figure 5C–E). According to the structural alignment of GDP and Mg^2+^ ions in the EV1–EV4 (Appendix A), the structures of GDP were aligned well in four energetic states, and they showed high stability throughout the entirety of the MR-GaMD simulations but, due to repulsive interactions of additional Na^+^ and Mg^2+^ions,they yielded evident deviations.

As for the GDP-bound C118S NRAS, MR-GaMD simulations captured three low energetic valleys EV1–EV3 (Figure 6A), verifying that the GDP-bound C118S NRAS is distributed at three conformational subspaces. The potential well EV2 was found to be deeper than that of EV1 and EV3; hence, the conformations of the GDP-bound C118S NRAS were mostly situated at EV2. The distances of D33 from Q61 were 12.6, 16.2, and 19.4 Å in the energy valleys EV1, EV2, and EV3 (Appendix A), respectively, suggesting that the switch domains SW I and SWII individually form a closed conformation in the EV1, an intermediate state in the EV2, and an open conformation in the EV3. According to Figure 6B–D, GDP not only produces salt bridge interactions with K16 and D119 but also forms HBIs with the common residues G13, V14, G15, K16, D30, N116, K117, S145, and K147. In addition, a new HBI is formed between GDP and A146 in EV1 (Figure 6B), three additional HBIs of GDP with S17, A18, and E31 are generated in EV2 (Figure 6C), and two new HBIs of GDP with S17 and A18 appear in EV3 (Figure 6D), which reflect the effect of conformational changes on interaction sites of GDP with NRAS. The representative structures in EV1–EV3 were superimposed together (Appendix A) and the results indicated that the switch domains SW I and SWII are highly disordered. In addition, the P-loop, the helix α3, and the loop L1 were found to generate slight deviations (Appendix A). Interestingly, two additional Na^+^ ions were detected in the EV2, which possibly affect the electrostatic environment around GDP (Appendix A). The structures of GDP and Mg^2+^ ions were aligned together (Appendix A). Although the adenine group and the middle ring of GDP were aligned well, the phosphate group of GDP and Mg^2+^ ions yielded evident deviations (Appendix A).

In summary, the analyses of the FELs revealed that G13D, Q61R, and C118S induce more energetic states than the GDP-bound WT NRAS, which increases the extent of disorder in the switch domains in the mutated states of NRAS. The work of Johnson et al. has verified that G13D leads to more open SWI and II and more disordered switch domains for NRAS due to the increased entropic contributions of conformations caused by G13D [31]. The representative conformations captured by the FELs indicated that G13D, Q61R, and C118S cause different numbers of Na^+^ ions to appear at some energetic states and compensate for the changes in the electrostatic environment induced by mutations, which certainly affect the interactions of GDP with NRAS. This interesting phenomenon was also found in previous work [77,78],supporting our current study well.

### 2.3. Principal Component Analysis

To better understand the effects of G13D, Q61R, and C118S on the conformational changes of the switch domains caused by NRAS, PCA was carried out using the SJMT. In general, eigenvalues arising from PCA are applied to clarify the total motion intensity of receptors. In the current study, the functions of eigenvalues as eigenvector indexes were used, as shown in Appendix A. The first several, larger eigenvalues mainly represent concerted motions of structural domains in NRAS. It can be seen from Appendix A that the first five eigenvalues account for 89.9, 92.4, 83.3, and 95.2% of the total movements of the GDP-bound WT, G13D, Q61R, and C118S NRAS, respectively. In comparison with the WT NRAS, the first eigenvalues of the GDP-bound G13D and C118S NRAS are increased, while those of the GDP-bound Q61R NRAS are greatly reduced. Therefore, G13D and C118S strengthen the total motion intensity of NRAS relative to the WT state but Q61R weakens it.

The first eigenvector corresponding to the first eigenvalue can reflect the main motion behavior of NRAS. Based on this, the first eigenvector stemming from the PCA was visualized using the software VMD [79], along with the optimized structure (Figure 7). The switch domains SW I and SW II have stronger motions compared to the other structural domains of NRAS. With regard to the GDP-bound WT NRAS, the two switches SWI and II move in an opposite direction away from each other, inducing a loose switch state; furthermore, the motion strength of SW I is stronger than that of the SW II (Figure 7A). By interacting with the WT NRAS, G13D changes the motion direction of SW I and SW II, and these two switches move close to each other (Figure 7B). Moreover, G13D not only alters the motion direction of the P-loop, the helix α3, and the loop L1 but also strengthens the movement of these three structural domains (Figure 7B). In contrast to the WT NRAS, Q61R changes the motion direction of the switch domains and inhibits their motion strength; at the same time, Q61R also strengthens the motion of loop L1 (Figure 7C). Compared to the WT NRAS, C118S slightly inhibits the movement of SW II and the partial regions in SW I, but it enhances the moving intensity of loop L1 and helix α3 (Figure 7D).

According to the above analyses, the three mutations G13D, Q61R, and C118S from NRAS generate different effects on the motion behavior of the switch domains. It is well-known that SW I is involved in the binding of NRAS to effectors; hence, G13D, Q61R, and C118S tune the activity of NRAS. Except for the obvious effect of G13D on the conformation of the P-loop, G13D, Q61R, and C118S also produce impacts on the movements of the helix α3 and the loop L1;structurally, these two regions are close to the allosteric position of NRAS to some extent. Thus, G13D, Q61R, and C118S possibly affect the allosteric regulation of the activity of NRAS. The conformational changes in the switch domain induced by the phosphorylation of Y32 disturb the binding of the RAS protein to effectors, which agrees with our results [80].

### 2.4. Interaction Networks of GDP and NRAS

The previous analysis of FELs indicates that conformational changes caused by G13D, Q61R, and C118S induce the alterations in the interaction sites for GDP and NRAS. To evaluate the stability of the GDP–residue interactions, the distances involved in the π–π interaction and the salt bridge interactions were calculated, and their frequency distributions are depicted in Figure 8. The occupancies of HBIs used to describe the stability of HBIs were analyzed using the program CPPTRAJ in Amber 20. The information on HBIs is listed in Table 1.

The structure of the lowest energy taken from the SJMT of the GDP-bound WT NRAS was utilized to depict the geometric position of the π–π interaction and the salt bridge interactions (Figure 8A). The distance of the salt bridge interaction between K16 and the phosphate group of GDP was calculated, and its frequency distribution is exhibited in Figure 8B. The distance of K16 from the phosphate group was distributed at 3.6, 3.4, 3.6, and 3.4 Å in the GDP-bound WT, G13D, Q61R, and C118S NRAS, respectively, suggesting that G13D and C118S strengthen the salt bridge interaction of K16 with GDP compared to the GDP-bound WT NRAS. The phenyl group of GDP generates the π–π interaction with the adenine group of GDP (Figure 8A), and the frequency distribution of its distance is displayed in Figure 8C. The peak value of the distance between the phenyl of F28 and the adenine group of GDP was located at 4.7 Å in the four current systems, and only the distribution width of the distance in the GDP-bound WT NRAS was wider than that in the three other systems; thus, G13D, Q61R, and C118S slightly strengthened the π–π interaction of F28 with GDP relative to the GDP-bound WT NRAS. The carbonyl of D119 produces the salt bridge interaction with the adenine group of GDP (Figure 8A), and the frequency distribution of this salt bridge distance is shown in Figure 8D. The peak value of the frequency distribution for this salt bridge was situated at 6.1 Å in the four current systems, implying that G13D, Q61R, and C118S hardly affect the strength of the salt bridge between D119 and the adenine group of GDP.

The structure of the lowest energy taken from the SJMT of the GDP-bound WT NRAS was also employed to describe the geometric position of HBIs (Figure 9). According to Table 1, except for V14, the residues G13(D13), G15, K16, S17, and A18, located at the P-loop, formed HBIs with the phosphate group of GDP in the four simulated systems (Figure 9A), and their occupancy was higher than 61.2% (Table 1), verifying that these HBIs were stable throughout the entirety of the MR-GaMD simulations. Compared to the GDP-bound WT NRAS, G13D, Q61R, and C118S increased the occupancies of HBIs of the phosphate group from GDP with G13(D13), G15, and K16 but decreased that of the phosphate group of GDP with A18 (Table 1). Although G13D increased the occupancy of the hydrogen bond between S17 and the phosphate group of GDP by interacting with the GDP-bound WT NRAS, Q61R and C118S reduced the occupancy of this hydrogen bond (Table 1). The residues N116 in loop L1 and S145, A146, and K147 in loop L4 yielded HBIs with the adenine group of GDP (Figure 9B,D) with an occupancy higher than 57.9% (Table 1), which suggests that these hydrogen bonds were stable during the MR-GaMD simulations. In contrast to the GDP-bound WT NRAS, G13D, Q61R, and C118S enhanced the occupancies of HBIs of the adenine group in GDP with N116 and A146 but decreased those of the adenine group in GDP with S145 and K147 (Table 1). The residues V29 and D30 from SW I and K117 in loop L1 formed HBIs with the middle ring of GDP and, in addition, D33 in SW I also produced an HBI with the middle ring of GDP in the GDP-bound G13D NRAS (Figure 9C,D); however, the occupancies of these HBIs were lower than 25.0%. These results suggest that the HBIs of V29 and D30 in the SW I with GDP are highly unstable. The aforementioned interaction sites of GDP and NRAS agree well with the GDP binding spots identified in the work of Johnson et al. [29].

Three conclusions can be drawn from the above results: (1) the salt bridge interaction of K16 and the HBIs of G13(D13), G15, K16, S17, and A18 with the phosphate group of GDP stabilize the conformation of the P-loop; (2) the salt bridge interaction of D119 and residues N116, S145, A146, and K147 with the adenine group of GDP lead to the ordering of the loops L1 and L4 compared to the switch domains; and (C) the instability in the HBIs of V29 and D30 with the middle ring of GDP is responsible for the structural disorder of the switch domains.

## 3. Materials and Methods

### 3.1. Initialization of Simulated Systems

The crystal structure taken from the Protein Data Bank (PDBID: 6ZIO) was used as the initial coordinates of the GDP-bound C118S NRAS [30]. To maintain the consistency of the atomic coordinates, the residue S118 in the 6ZIO was directly revised as C118 to generate the GDP-bound WT NRAS. Then, the G13 and Q61 in the GDP-bound WT NRAS were separately changed into D13 and R61 to yield the GDP-bound G13D and Q61R NRAS by using the Leap module in Amber 20 [81,82]. The Mg^2+^ ion in the crystal structure was kept in the starting models of the four NRAS-related systems. The program H++ [83] was used to check and assign rational protonation states to the residues in NRAS. The chemical bonds between the missing hydrogen atoms in the crystal structure and the heavy atoms were constructed using the Leap module. The force field parameters of NRAS were obtained using ff19SB force field [84]. The force field parameters of GDP were taken from the work of Meagher et al. [85]. An octahedral periodic box of water with a buffer of 12.0 Å was adopted to solve each NRAS-related complex and the force field parameters of water molecules were assigned through the TIP3P model [86]. The appropriate numbers of sodium ions (Na^+^) were placed around each complex in the 0.15 M NaCl salt environment to produce a neutral simulation system, in which the parameters of Na^+^, Cl^−^, and Mg^2+^ were obtained from the Aqvist force field [87].

### 3.2. MR-GaMD Simulations

To obtain data with regard to the conformational changes of NRAS, MR-GaMD simulations are executed with the GDP-bound WT, G13D, Q61R, and C118S NRAS. All of the systems firstly underwent40,000step steepest descent minimization and then minimization using a3000step conjugate gradient, which was utilized to remove bad atom–atom contacts. Subsequently, the temperature of the four systems was enhanced from 0 to 310 K through a soft heating procedure within 1 ns in the NVT condition, in which non-hydrogen atoms were constrained by means of a weak harmonic restriction of 2 kcal·mol^−1^·Å^−^^2^.After that, a 2ns equilibrium of 310 K was implemented to deeply relax the four systems in the NPT ensemble and a restriction similar to that for the heating process was also adopted. Finally, a cMD simulation was run for 100 ns to analyze the time evolution of the four systems at a constant temperature of 310 K and pressure of 1 bar, employing periodic boundary conditions and the particle mesh Ewald (PME) method [88,89]. Two new conformations randomly extracted from the previous 100ns cMD simulation were utilized as the initial coordinates to restart two new cMD simulations by randomly assigning initial atomic velocities for each conformation with the Maxwell distribution. The final structures obtained from the above three replica cMD simulations were adopted to run MR-GaMD simulations.

In this work, the GaMD simulations employed the harmonic boost potential to reduce the free energy barriers of the systems [60], which greatly enhanced the conformational sampling of the NRAS-related systems. The system potential $V\left(\vec{r}\right)$ was updated as $V^{*}\left(\vec{r}\right)$ if $V\left(\vec{r}\right)$ was lower than a threshold energy E, which is clarified in Equation (1):(1)$$V^{*}\left(\vec{r}\right)=V\left(\vec{r}\right)+\Delta V\left(\vec{r}\right)$$
(2)$$\Delta V\left(\vec{r}\right)=\left\{\begin{array}{c} 0,\;\&\;V\left(\vec{r}\right)\geq E\\ \frac{1}{2}k{\left(E-V\left(\vec{r}\right)\right)}^{2},\;\&V\left(\vec{r}\right)<E \end{array}\right.$$
in which *k* denotes the harmonic force constant, and the two parameters *E* and *k* may be adjusted with the help of the three enhanced sampling principles described in Equations (3) and (4):(3)$$V_{max}\leq E\leq V_{min}+\frac{1}{k}$$
(4)$$k=k_{0}\frac{1}{V_{max-V_{min}}}$$

Ifa lower bound $E=V_{max}$ is assigned to E, then the parameter $k_{0}$ is generatedusing Equation (5):(5)$$k_{0}=\min\left(1.0,\;\frac{\sigma_{0}}{\sigma_{V}}\Delta \frac{V_{max}-V_{min}}{V_{max}-V_{avg}}\right)$$

Inversely, if an upper bound $E=V_{min}+\frac{1}{k}$ is given to E,then the parameter $k_{0}$ is obtainedthrough Equation (6):(6)$$k_{0}=\left(1.0-\frac{\sigma_{0}}{\sigma_{V}}\right)\Delta \left(\frac{V_{max}-V_{min}}{V_{avg}-V_{min}}\right)$$
in which $V_{max}$, $V_{min}$, and $V_{avg}$ represent the minimum, maximum, and averaged potential energies of the systems, respectively. The parameter $\sigma_{V}$ indicates the standard deviation of the system potential energies, while the parameter $\sigma_{0}$ is a user-specified upper limit for accurately reweighting. For this work, 3.6 µs MR-GaMD simulations, composed of three separate GaMD simulations of 1.2 μs, were implemented with the GDP-bound WT, G13D, Q61R, and C118S NRAS. Three replica GaMD trajectories were connected into a single-joined MR-GaMD trajectory (SJMT) so as to be convenient for the post-process analysis. The PyReweighting toolkit provided by Miao et al. was used to reweight the data from the post-processing analysis and to detect the original energetic profiles of the four NRAS-related systems [90]. As for all cMD and GaMD simulations, all chemical bonds between hydrogen atoms and heavy ones obeyed a restriction set by the SHAKE algorithm [91]. The temperatures of the four systems were regulated with a Langevin thermostat, in which a collision frequency of 2.0 ps^−1^ was adopted [92]. The PME method with an appropriate cutoff value of 10 Å was adopted for the estimation of non-bond interactions. The module pmemd.cuda included in Amber 20 was employed to run all simulations [93,94].

### 3.3. Principal Component Analysis

PCA was conducted to explore concerted motion couplings with functional significance in this work. The first step in the PCA was to build a covariance matrix C of the atoms Cα by using Equation (7):(7)$$C=\langle \left(q_{i}-\langle q_{i}\rangle \right){\left(q_{j}-\langle q_{j}\rangle \right)}^{T}\rangle$$
where $q_{i}$ and $q_{j},$ respectively, signify the Cartesian coordinates of the *i*th and *j*th$\;C_{\alpha }$ atoms fromNRAS, while $\langle q_{i}\rangle$ and $\langle q_{j}\rangle$ indicate their averaged positions over conformational ensembles. In general, the average is estimated by superimposing the SJMT with a defined referenced structure to remove overall translations and rotations through a least-square fit procedure [95]. The second step of the PCA was to perform diagonalization on the symmetric matrix C to yield a diagonal one A by means of an orthogonal coordinate transformation matrix T based on the following equation:(8)$$A=T^{T}C_{ij}T$$
in which the diagonal elements of A are the eigenvalues $\lambda_{i}$ and the columns of A correspond to the eigenvectors reflecting the motion direction relative to $\langle q_{i}\rangle$. The third step of the PCA was to explore the concerted movements of the structural domains in a multidimensional space using the eigenvector and to clarify the motion strength along an eigenvector, which could rationally reflect the conformational changes in structural domains from NRAS. The PCA was implemented by utilizing the CPPTRAJ module in Amber 20 [96]. The software VMD was employed to realize the visualization of the data generated by the PCA [79], plot pictures, and unveil the impacts of G13D, Q61R, and C118S on the conformational alterations of NRAS.

### 3.4. Dynamics Cross-Correlation Maps

To check the internal dynamics of the structural domains from NRAS, the elements $C_{ij}$ of DCCMs were calculated by using the x , y, and z coordinates of the $C_{\alpha }$ atoms from NRAS with Equation (9):(9)$$C_{ij}=\frac{\langle \Delta r_{i}\Delta r_{j}\rangle }{{\left(\langle \Delta r_{i}^{2}\rangle \langle \Delta r_{j}^{2}\rangle \right)}^{1/2}}$$
where $\Delta r_{i}$ and $\Delta r_{j}$ respectively indicate the displacement of atoms i and j relative to their averaged positions, while the angle brackets describe ensemble averages over the conformations saved at the SJMT. The element values of DCCMs fluctuated in a range from −1 to 1. The $C_{ij}$ of the positive and negative values characterized the positively correlated motions and the anti-correlated movements between atoms i and j, respectively. Color-coded patterns were employed to represent the extent of the correlated motions. The module CPPTRAJ in Amber 20 was utilized to perform the calculations of DCCMs.

## 4. Conclusions

NRAS is regarded as an important target in drug design for anticancer treatment. Mutation-mediated conformational transformation of the switch domains can induce alterations in the activity of NRAS. In this work, 3.6 μs MR-GaMD simulations, consisting of MR-GaMD simulations withthreeseparate1.2 μs replications, were performed on the GDP-bound WT, G13D, Q61R, and C118S NRAS to enhance conformational samplings and decipher the effects of mutations on the activity of NRAS. The analyses of the RMSFs and MSAs indicated that G13D, Q61R, and C118S not only change the structural flexibility of the switch domains but also reduce the global structural flexibility of NRAS. The DCMM calculations suggest that three mutations alter the correlated motions of the switch domains relative to the P-loop from the GDP-associated NRAS. PCA performed with the SJMT showed that the motion behavior of the switch domains from the GDP-bound NRAS was strongly disturbed. The switch domains are involved in binding domains of NRAS to effectors; hence, the alterations in the structural flexibility and motion behavior of the switch domains in NRAS produce certain influences on the binding of NRAS to effectors.

The FELs constructed by using the distance of D33 from residue 61 and the RMSDs of backbone atoms verified that G13D, Q61R, and C118S induce more energetic states in the GDP-associated mutated NRAS relative to the GDP-bound WT NRAS, which results in the switch domains in the GDP-bound mutated NRAS attaining a more disordered state than in the GDP-bound WT one. The switch domains are responsible for the NRAS—effector binding; thus, the changes in the degrees of order of the switch domains certainly regulate the activity of NRAS. The analyses of the GDP–residue interactions verified that the instabilities in the HBIs of GDP with V29 and D30 in SWI drive the high disorder of the switch domains in NRAS, which plays an important role in regulations of the activity of NRAS. We expect that this study can contribute useful information for understanding the target role of NRAS in drug development for anticancer treatment.

## Figures and Tables

**Figure 1 molecules-27-05596-f001:**
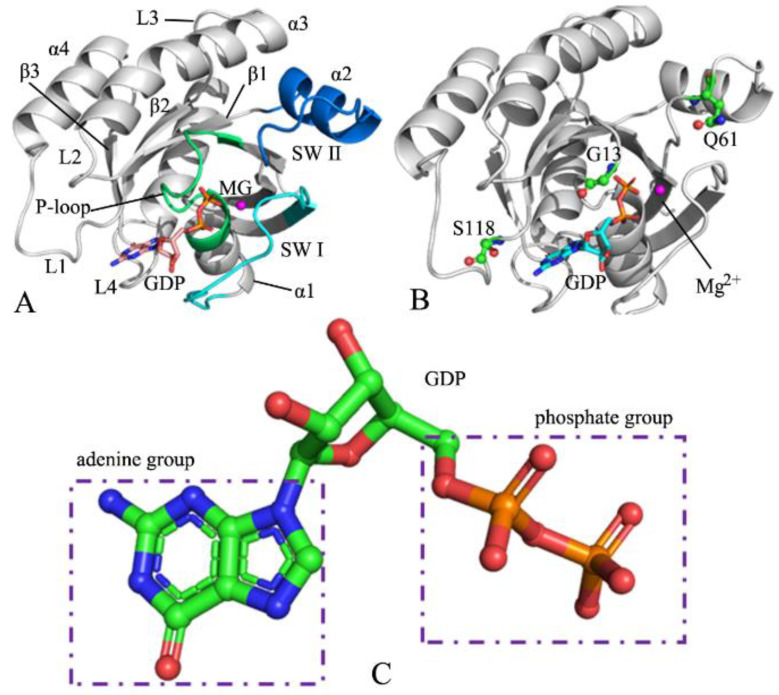
Molecular structures: (**A**) the GDP-bound NRAS, in which NRAS is displayed in cartoon style and GDP in stick style; (**B**) mutation sites, in which mutated residues are exhibited in the ball and stick style; and (**C**) GDP shown in the ball and stick style.

**Figure 2 molecules-27-05596-f002:**
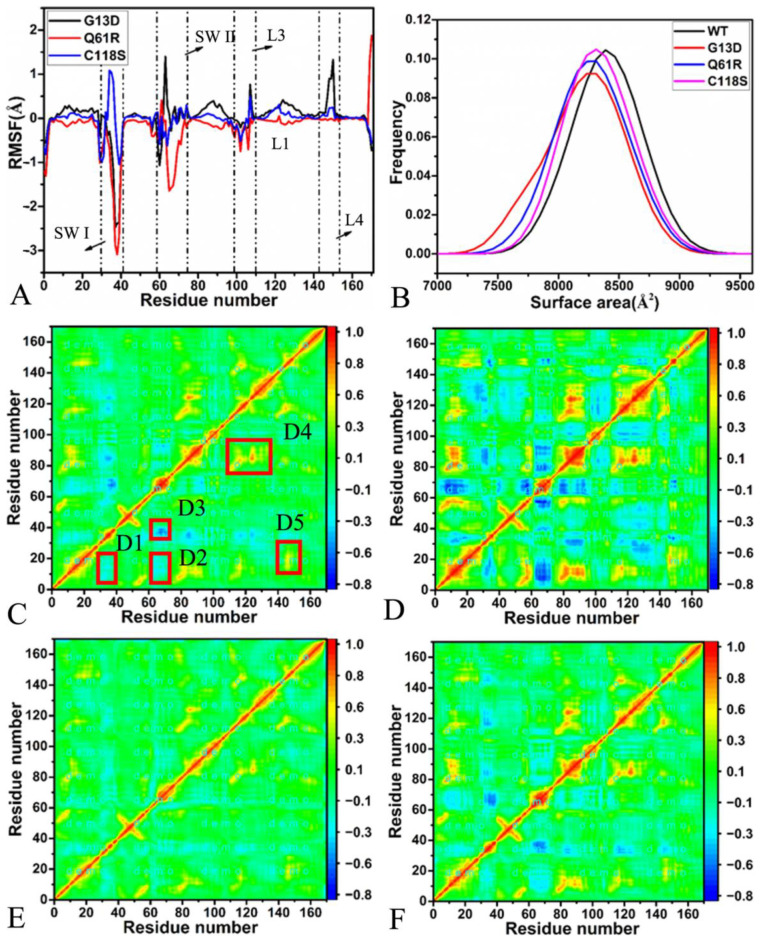
Structural flexibility and motion modes. (**A**) The differences in RMSFs forthe atom C_α_ between the mutated and WT NRAS, (**B**) the molecular surface areas of the WT and mutated NRAS, (**C**) the dynamics cross-correlation map (DCCM)of the GDP-bound WT NRAS, (**D**) the DCCM of the GDP-bound G13D NRAS, (**E**) the DCCM of the GDP-bound Q61R NRAS, and (**F**) the DCCM of the GDP-bound C118S NRAS.

**Figure 3 molecules-27-05596-f003:**
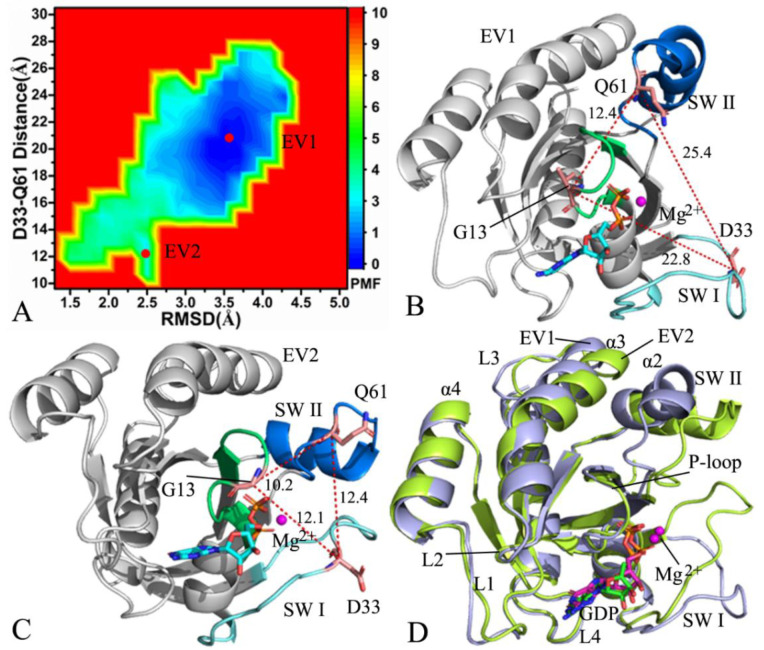
FELs and representative conformations of the GDP-bound WT NRAS: (**A**) FELs constructed by using the distance of D33 from Q61 and the RMSDs of backbone atoms as reaction coordinates, (**B**) the conformation located at the energy valley EV1, (**C**) the conformation situated at the energy valley EV2, and (**D**) superimposition of structures in EV1 and EV2.

**Figure 4 molecules-27-05596-f004:**
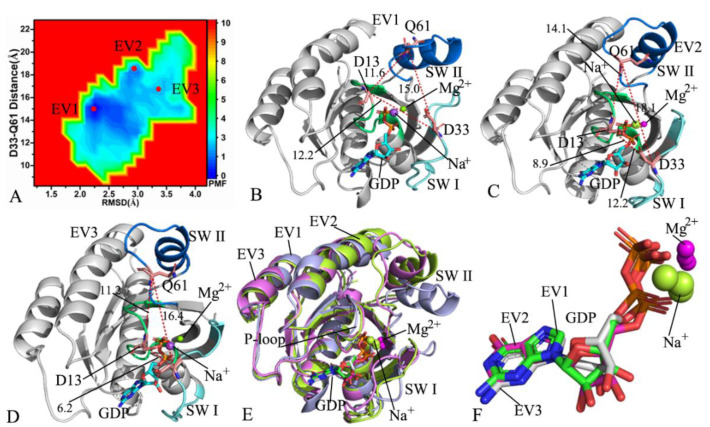
FELs and representative conformations of the GDP-bound G13D NRAS: (**A**) FELs constructed by using the distance of D33 from Q61 and the RMSDs of backbone atoms as reaction coordinates; (**B**–**D**) the representative conformations located at the energy valleys EV1, EV2, and EV3, respectively; (**E**) superimposition of representative structures in EV1, EV2, and EV3; and (**F**) superimposition of GDP and Mg^2+^ ion in EV1, EV2, and EV3.

**Figure 5 molecules-27-05596-f005:**
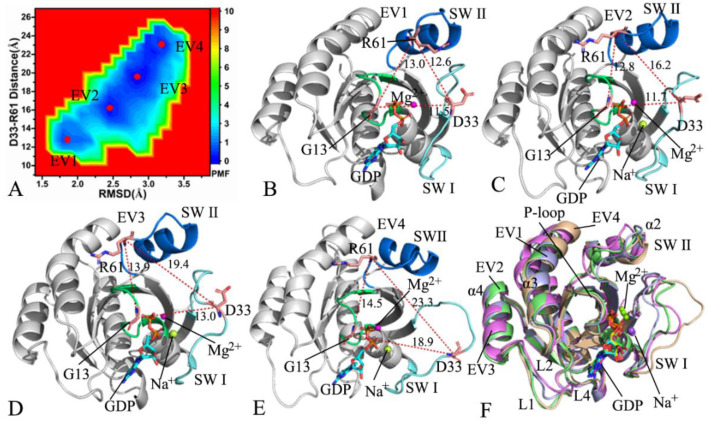
FELs and representative conformations of the GDP-bound Q61R NRAS: (**A**) FELs constructed by using the distance of D33 from R61 and the RMSDs of backbone atoms as reaction coordinates; (**B**–**E**) the conformations located at the energy valleys EV1, EV2, EV3, and EV4, respectively; (**F**) superimposition of representative structures in EV1, EV2, EV3, and EV4.

**Figure 6 molecules-27-05596-f006:**
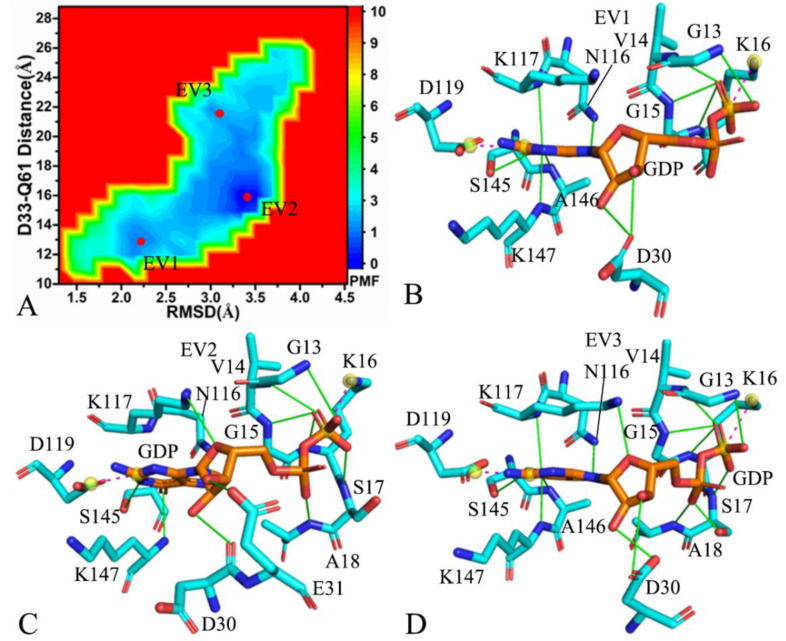
FELs and interaction sites from representative conformations of the GDP-bound C118S NRAS: (**A**) FELs constructed by using the distance of D33 from Q61 and the RMSDs of backbone atoms as reaction coordinates, (**B**–**D**) interaction sites of representative conformations in the energy valleys EV1, EV2, and EV3.

**Figure 7 molecules-27-05596-f007:**
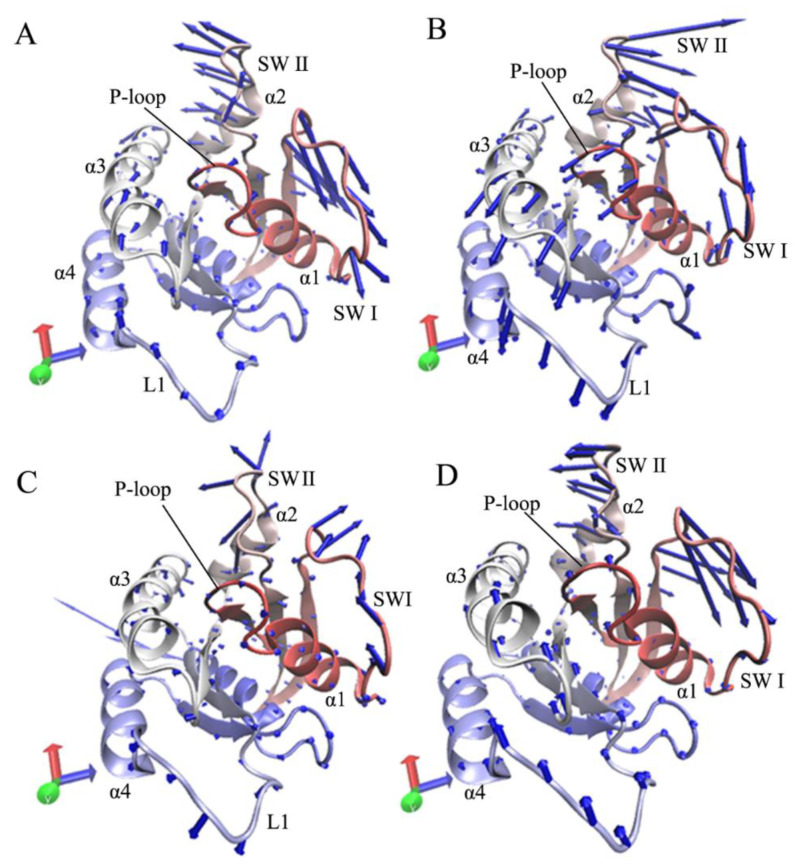
Concerted motions of structural domains from NRAS revealed by the first eigenvector from the PCA: (**A**) the GDP-bound WT NRAS, (**B**) the GDP-bound G13D NRAS, (**C**) the GDP-bound Q61R NRAS, and (**D**) the GDP-bound C118S NRAS.

**Figure 8 molecules-27-05596-f008:**
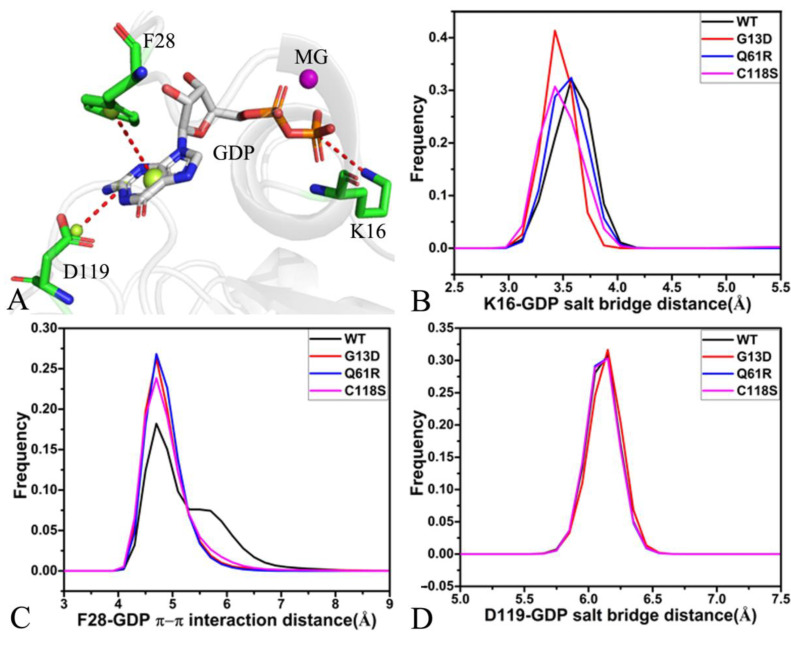
Structures and interactions of GDP and NRAS: (**A**) geometric positions of key interactions, (**B**) the frequency distribution of the distance for the salt bridge interaction between K16 and the phosphate group of GDP, (**C**) the frequency distribution of the distance for the π–π stacking interaction of F28 with the adenine group of GDP, and (**D**) the frequency distribution of the distance for the salt bridge interaction between D119 and the adenine group of GDP.

**Figure 9 molecules-27-05596-f009:**
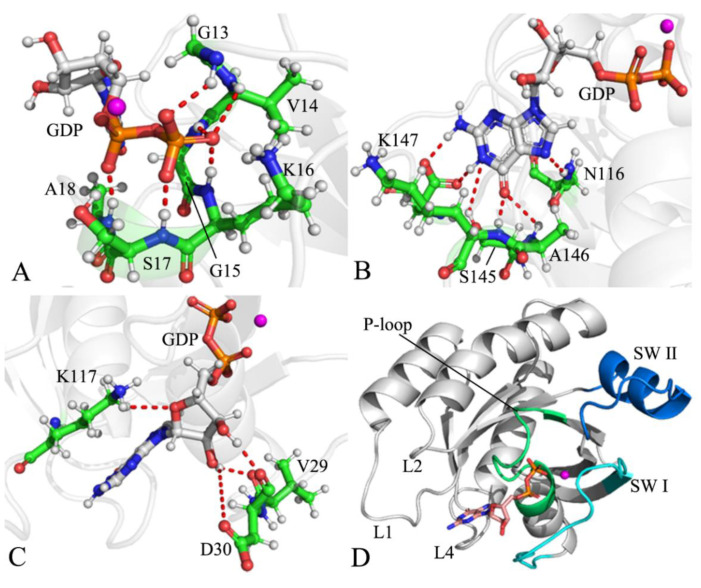
HBIs of GDP with NRAS: (**A**) HBIs of the phosphate groups in GDP with residues of NRAS, (**B**) HBIs of the adenine group in GDP with residues of NRAS, (**C**) HBIs of the middle ring in GDP with residues of NRAS, and (**D**) the structural regions of NRAS involved in HBIs.

**Table 1 molecules-27-05596-t001:** Hydrogen bonding interactions of GDP with NRAS analyzed by the program CPPTRAJ.

Hydrogen Bonds ^a^		Occupancy(%) ^b^			
Residue	GDP	WT	G13D	Q61R	C118S
K16-N-H	O1B	97.5	99.9	99.1	98.1
A18-N-H	O1A	84.6	79.4	72.8	69.6
G15-N-H	O1B	97.4	98.7	98.5	99.2
S17-N-H	O2B	84.6	97.2	73.5	69.6
#13-N-H ^c^	O3B	74.3	94.8	85.1	88.8
V14-N-H	O1B	14.5	16.3	15.4	16.2
N116-ND2HD21	N7	86.5	91.2	87.8	88.5
S145-OG-HG	N1	60.8	57.9	57.9	58.4
K147-N-H	O6	87.9	59.9	84.9	83.1
G15-N-H	O3A	61.2	74.6	65.2	66.1
A146-N-H	O6	59.7	73.7	65.3	64.1
D119-OD1	N1-H1N	84.5	96.3	87.6	89.1
D119-OD2	N2-H21	74.8	94.8	79.1	79.4
K117-NZ-HZ2	O4′	23.2	30.5	29.6	28.7
V29-O	O2′-H2′	11.1	19.5	17.4	24.7
D30-OD1	O2′-H2′	17.2	14.4	23.4	22.2
D30-O	O3′-H3′	15.9	13.1	21.9	20.8
D33-OD1	O3′-H7′	0.0	12.1	0.0	0.0

^a^ Hydrogen bonding interactions were recognized by an acceptor–donor distance of <3.5 Å and acceptor–H-donor angle of >120°. ^b^ Occupancy (%) is defined as the percentage of simulation time that a specific hydrogen bond existed. ^c^ The symbol #13 represents the residues G13 and D13.

## Data Availability

Not applicable.

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
