# Peer review of "Deciphering Conformational Changes of the GDP-Bound NRAS Induced by Mutations G13D, Q61R, and C118S through Gaussian Accelerated Molecular Dynamic Simulations"

_molecules, 2022, doi:10.3390/molecules27175596_

Round 1

Reviewer 1 Report

The abstract is not very precised, according the way of dealing with terminology about the MD.

The conformational changes of the switch domains significantly affect the activity of NRAS. - No information about the switch domain before. Add and rewrite the sentence.

 Dynamics analyses show that the structural flexibility, motion modes, (comma)
and dynamic(s - remove) behavior of the switch domains involved in binding of NRAS to effectors are (IS)  highly altered by G13D, Q61R, (comma) and C118S, which in turn regulates the activity of NRAS. - the way of trying to explain the following was a bit odd, the way of starting the sentence. Dynamic? What dynamic? More details about and careful use of dynamics in the case of that paper.

The information from free energy landscapes (FELs) suggests that G13D, Q61R and C118S induce more energe... - Information? We are not getting information from Free Energy, why you are using the Free Energy?

Missing the ref. for the molecular structures of GTP  determined by X-ray and NMR experiments from different work groups, please have a look G. Pálfy , I. Vida and A. Perczel , Biomol. NMR Assignments, 2020, 14 , 1 —7

2.1 structural flexibility and international dynamics of NRAS  - are you shoure about the subtitle of the 2.1?

The way of presenting the 2.1 very confusing for the reader.

Provide the change of the Hydrogen bonding over the time, as well SASA and comment.

Also, commenet about the hydrophobisity of the Amino acids.

About the cMD simulation is run for 100 ns to perform time evolution of four system at constant  - very short MD time, cant accept the results so of the obtained results with 100 ns. Please, prolonge

SJMT - missing any information and ref. in the results section?

About the frequency distribution plots - explain how was performed?

Please add a plot about the change in conformational of the domain from its proactive confirmation to the active state identified through GaMD.

Reviewer 2 Report

This is a nice paper on molecular dynamics (MD) simulations of GDP bound NRAS proteins with mutations.

There are some minor changes. Make it clear that the AMBER code was used for the MD simulations.

Define all abbreviations when first used in the title, abstract, text.

In table 1 and the text, I doubt that the values are good to 0.1%. They are really going to be good to 1%.

In the conclusions, make it clear what the conclusions about mechanism are separate from summarizing the approach. I would use at least 2 paragraphs here to make this clear and to expand on the mechanistic conclusions.

Please fix the author contributions section which has no names.

Round 2

Reviewer 1 Report

I think that the work is ready  to be published.